# Synthesis of precisely functionalizable curved nanographenes via graphitization-induced regioselective chlorination in a mechanochemical Scholl Reaction

Jovana Stanojkovic [1], Ronny William[1], Zhongbo Zhang [1], Israel Fernández [2], Jingsong Zhou[1], Richard D. Webster[1] & Mihaiela C. Stuparu [1] ✉

While the synthesis of nanographenes has advanced greatly in the past few years, development of their atomically precise functionalization strategies remains rare. The ability to modify the carbon scaffold translates to controlling, adjusting, and adapting molecular properties. Towards this end, here, we show that mechanochemistry is capable of transforming graphitization precursors directly into chlorinated curved nanographenes through a Scholl reaction. The halogenation occurs in a regioselective, high-yielding, and general manner. Density Functional Theory (DFT) calculations suggest that graphitization activates specific edge-positions for chlorination. The chlorine atoms allow for precise chemical modification of the nanographenes through a Suzuki or a nucleophilic aromatic substitution reaction. The edge modification enables modulation of material properties. Among the molecules prepared, corannulene-coronene hybrids and laterally fully π-extended helicenes, heptabenzo[5]superhelicenes, are particularly noteworthy.

Nanographenes are a fascinating class of molecules with potential in materials science[1]. Thus, in recent years, great progress has been made in developing rational synthesis of such macromolecules[2–4]. A majority of work, however, relates to planar structures. In comparison, the development of general synthetic methods towards curved structures still remains challenging[5–12]. Even more demanding is the task of addressing these structures in a precise manner for further scaffold functionalization. Such functionalization opportunity could be crucial in adapting material properties such as redox potentials or solubility to meet a certain requirement for a desired application. Typically, nanographene precursors are modified in a multi-step synthetic procedure to gain entry to a certain functionalized structure. Recently, post-synthesis modification of the nanographene edge has emerged as a valuable alternative. In this approach, the nanographene is first synthesized and then activated for further modification. For instance, substitution of nanographene edge with borylation[13–15] or perchlorination[16,17] allows for installation of functional groups through transition-metal-catalyzed coupling reactions. Despite these advances, selective edge functionalization approaches that can occur in a general and efficient manner remain unknown.

In this work, we report a general and high-yielding synthetic route to non-planar nanographenes with precisely functionalizable scaffolds. In our approach, the edge-chlorination occurs within the same step as the nanographene formation. Therefore, nanographenes do not need to be subjected to the activation step separately and can be directly subjected to the functionalization step after their preparation. Furthermore, the edge-chlorination is selective and thus functionalization occurs in a regioselective manner. Finally, unlike the aforementioned solution-phase reactions, the current strategy relies on an environmentally-friendly mechanochemical (solvent-less) reaction[18–24].

[1]School of Chemistry, Chemical Engineering and Biotechnology, Nanyang Technological University, 21 Nanyang Link, 637371 Singapore, Singapore. [2]Departamento de Química Orgánica I, Centro de Innovación en Química Avanzada (ORFEO-CINQA), Facultad de Ciencias Químicas, Universidad Complutense de Madrid, 28040 Madrid, Spain. ✉e-mail: mstuparu@ntu.edu.sg

In this strategy, alkyne-carbonyl metathesis is first employed to give a key corannulene-based indanone molecule. This species is capable of being transformed into a cyclopentadienone and combined with a variety of aryl alkynes and arynes to form graphitization precursors through a Diels-Alder reaction. In these precursors, corannulene represents an electron-deficient curved fragment whereas the Diels-Alder adduct represents an electron-rich fragment. A mechanochemical Scholl reaction then planarizes the adduct and simultaneously annulates it to the curved fragment. This process is referred to as 'graphitization', inspired by the earlier use of this term by Borchardt and co-workers in the context of coronene synthesis[25]. The graphitization creates a large overall curved molecule. A solution-phase Scholl reaction, on the other hand, leads to an intractable mixture of compounds. Remarkably, the mechanochemical process of graphitization activates the curved fragment for chlorination in such a manner that only the adjacent carbon atoms to the annulated structure become chlorinated. Thus, nanographenes can be chemically modified in a regioselective manner. The modifications allow for tuning the electronic properties.

Overall, the developed synthesis is general, efficient, and reproducible. Among the various structures prepared, two types of structures are particularly noteworthy. One in which the smallest non-planar and planar graphene motifs, corannulene[26,27] and coronene[28], are combined with a maximum of π-overlap and share one benzene ring between them. If the planar fragment is viewed as hexabenzocoronene[29], then the two structures share phenanthrene-like three aromatic rings between them. Despite efforts to prepare hybrids of these archetypical motifs in nanographene chemistry[30], such an extent of structural overlap is exceptional and allows for documenting the properties of such molecular hybrids. The other is an example of a [5]helicene structure that is completely embedded in a π-envelope. This enhances the repertoire of the laterally fully π-extended superhelicenes[31]. The members of this material class are rare[32–34]. Overall, therefore, this work harnesses graphitization-induced selective edge chlorination in a mechanochemical Scholl reaction to establish an efficient synthesis of curved nanographenes with unique structures and a precisely functionalizable scaffold which allows for tuning of the molecular properties.

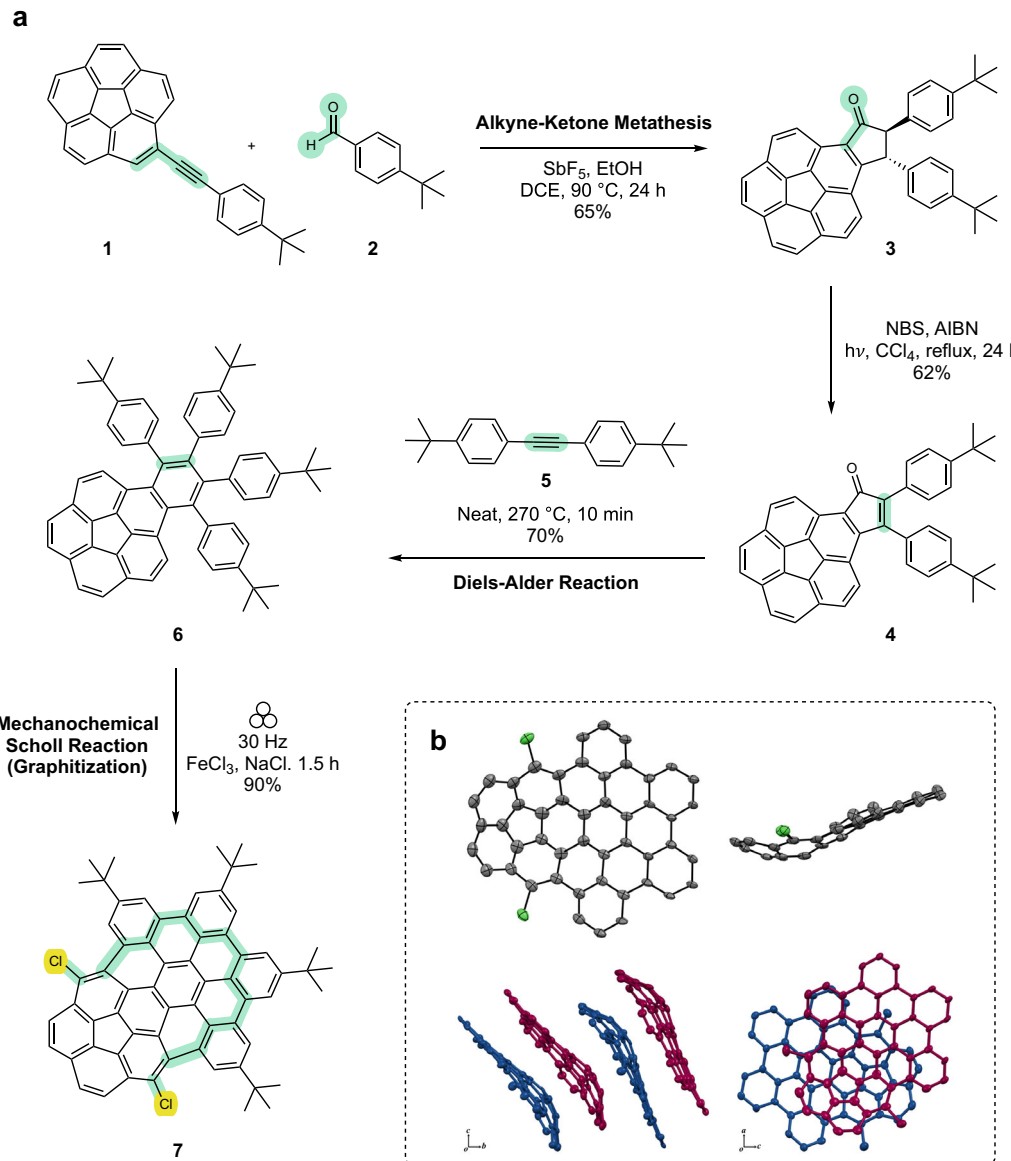

**Fig. 1 | Synthesis and characterization. a** Reaction sequence to obtain regioselectively chlorinated nanographenes. Reactive functional groups and newly formed bonds are highlighted. **b** X-Ray crystal structure of **7**. Thermal ellipsoids were scaled at 50% probability level. *tert*-Butyl groups and hydrogen atoms were omitted for clarity. Top view, side view, and packing structures are shown in a clockwise fashion.

## Results and discussion

### Synthesis

In the presence of catalytic Lewis or Brønsted acids, the alkyne-carbonyl metathesis[35] provides an efficient access to α,β-unsaturated carbonyl compounds. In the present context, corannulene-based phenylene ethynylene **1** and aldehyde **2** were chosen as the reactants (Fig. 1a). The reaction was carried out with a catalytic amount of SbF$_5$ with ethanol as an additive to efficiently convert the reactants into 2,3-disubstituted indanone **3** in a one-pot procedure. Subsequently, radical benzylic bromination and elimination of HBr in one step leads to the formation of cyclopentadienone **4** (Supplementary Fig. 1). A Diels−Alder cycloaddition between **4** and alkyne **5** yields graphitization precursor **6** under neat conditions (Supplementary Fig. 2).

Recently, mechanochemistry has emerged as a potent synthetic tool[18–24]. In this approach, mechanical energy through grinding or milling action is used to promote a chemical reaction[36]. It is sustainable[37] with potential for altering chemical reactivity and product selectivity[21,38]. The Scholl reaction has been particularly successful under mechanochemical conditions[39–42]. Thus, precursor **6** was subjected to an iron(III) chloride-mediated mechanochemical Scholl reaction to furnish nanographene **7**[43]. The solvent-less reaction was carried out in a mixer mill equipped with a ZrO$_2$ grinder jar and a ZrO$_2$ ball. Sodium chloride was used as a milling auxiliary. The reaction was milled for 1.5 h at a frequency of 30 Hz. A MALDI-TOF mass spectrometry characterization indicated successful graphitization albeit with bis-chlorination (see Supplementary Information for characterization details). The concrete structural evidence came from the single crystal X-ray diffraction analysis which confirmed the number and position of the chlorine atoms on the nanographene scaffold (Fig. 1b). The isolated yield of **7** was consistently found to be ~90% over several reproductions.

In the Scholl reaction, typically, chlorination of the aromatic scaffold is an undesired side reaction leading to minor products which are removed during purifications. It is also uncontrolled. For instance, in a recent work by Borchardt and co-workers, coronene is chlorinated in a mechanochemical Scholl reaction to give a mixture of products containing 1–18 chlorine atoms[44]. On the other hand, if chlorination is desired, chlorine atoms are introduced after the synthesis of

**Fig. 2 | Modularity of synthesis. a** Mono- and bis-chlorination strategy by employing a free acetylene reactant. **b** Application of arynes as coupling partners.

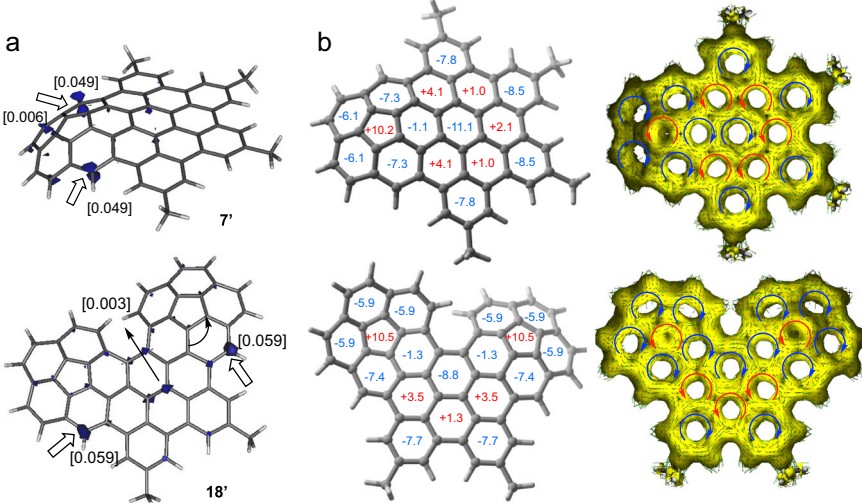

**Fig. 3 | Origin of regioselectivity and aromaticity. a** Plot of the Fukui's susceptibilities (isosurface value of 0.025 au) for **7′** (top) and **18′** (bottom). Values in brackets indicate the corresponding computed Fukui functions. **b** AICD plots (right) computed for the nanographenes (isovalue of 0.035 au). Clockwise (blue) and anti-clockwise (red) arrows indicate diatropic (aromatic) and paratropic (antiaromatic) ring currents. NICS values (in ppm) are shown in the optimized structures (left). Negative (aromatic) values are given in blue whereas positive (antiaromatic) values are given in red.

nanographenes (in a post-synthesis modification step) under harsh and toxic conditions with solvents such as CCl₄, elevated temperatures, and long reaction times[16]. Such procedures culminate into total chlorination (perchlorination) of the molecules. Therefore, the observation of a regioselective bis-chlorination adjacent to the positions where the planar and electron-rich fragment annulates to the non-planar and electron-poor fragment was remarkable. Hence, a new series of graphitization precursors were planned to further probe this aspect (Fig. 2a). Through the use of a terminal phenyl acetylene **8**, we obtained a 1:2 mixture of two isomers **9** and **10** (Supplementary Fig. 3) which could be separated easily by column chromatography. These isomers differ in the position of the phenyl rings. Upon a Scholl reaction, **9** would lead to only one connection between the planar and the non-planar fragments while **10** would give two annulations exactly as in the case of nanographene **7**. If the regioselectivity originates from activation of the corannulene fragment by the planar fragment, **9** should lead only to mono-chlorination and **10** should give the previously observed bis-chlorination. Indeed, **9** and **10** form mono- and bis-chlorinated compounds **11** and **12**, respectively, after the Scholl reaction.

To test modularity of the synthesis as well as the general nature of regioselective chlorination, alkyne precursors were replaced by two aryne precursors in the Diels-Alder reaction (Fig. 2b). In these cases, the aryne, due to its high reactivity, was generated in-situ in a microwave-assisted fluoride-induced 1,2-elimination of *ortho*-silylaryltriflates reaction[45]. In one case, benzyne **13** was employed whereas corannulyne **14** was used in the other[46]. **13** led to the synthesis of **15** (Supplementary Fig. 4) with a naphthalene nucleus while **14** led to the formation of a bis-corannulene system **17**. A mechanochemical Scholl reaction then produced **16** and **18**, respectively from precursors **15** and **17**. Consistent with the previous results, mono-chlorination occurred in **16** (Supplementary Fig. 5) and bis-chlorination in **18**. Once again, despite the presence of the naphthalene unit, the chlorination occurred only adjacent to the aromatic fusion place on the corannulene nucleus. These results confirm the regioselective nature of chlorination and the generality of the synthesis.

## Regioselectivity and aromaticity

Intrigued by the chlorination results, a computational study at the dispersion corrected B3LYP-D3/def2-SVP level was carried out to gain more insight into the regioselectivity of the process. To this end, we computed the preferred chlorination sites in the representative model precursors **7′** and **18′** (leading to the observed chlorinated compounds **7** and **18**, respectively, where the *t*-Bu groups were replaced by methyl groups) by means of the corresponding condensed Fukui functions $(f_k^-)$[47]. These reactivity indicators were chosen because they have been successfully used in the past to rationalize the regioselectivity of related electrophilic substitutions including chlorination and bromination reactions[48,49]. As depicted in Fig. 3a, the positions adjacent to the newly formed bonds between the planar and the curved corannulene fragments exhibit the highest $f_k^-$ values and therefore are the most susceptible positions to be chlorinated, which is consistent with the experimental findings.

To further gain understanding of these results, the aromatic nature of the newly prepared systems and their associated ring currents were studied[50–52]. For this, we first computed the variation of the Nuclear Independent Chemical Shift (NICS)[53] values of the different rings in going from the parent cyclopentadienone **4** to model nanographenes **7′** and **18′**. As graphically shown in Fig. 3b and Supplementary Fig. 6, the aromaticity of the rings belonging to the corannulene moiety in **4** remains essentially unaltered upon the fusion in the corresponding nanographenes. Interestingly, in all cases, whereas the central six-membered ring fused to the corannulene fragment (i.e. that formed during the Diels-Alder reaction) is aromatic, the six-membered rings fused to the edge corannulene rings (i.e. those formed during the Scholl reaction) become clearly antiaromatic. This is consistent with the Anisotropy of the Induced Current Density (AICD)[54,55] method, which nicely confirms the occurrence of a paratropic (i.e. antiaromatic) ring current in these fused six-membered rings (Fig. 3b). Strikingly, these antiaromatic rings are fused to the six-membered rings of the corannulene where the chlorination takes place, therefore suggesting a strong correlation between aromaticity and reactivity.

## Solution-phase Scholl reaction

At this point, we became interested in a comparison of the mechanochemical reaction with the traditional solution-phase reaction. For this, the graphitization reaction of the precursor **6** was carried out under typical Scholl conditions using DDQ/TfOH, DDQ/CH₃SO₃H, and FeCl₃ in dichloromethane as a solvent. Unfortunately, the solution-phase reaction reproducibly produced an intractable reaction mixture. In ¹H NMR spectroscopy, the crude spectrum showed a number of

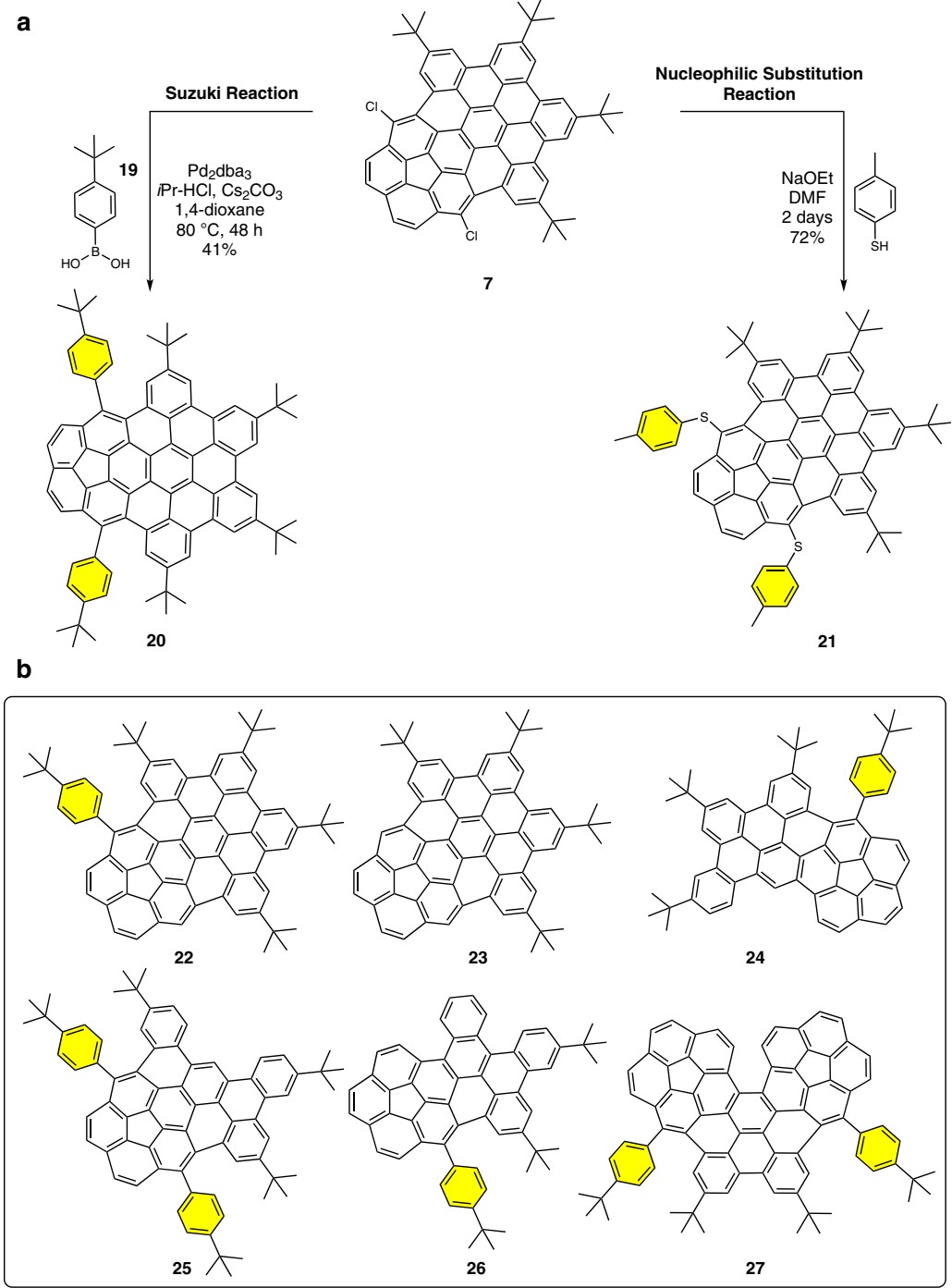

**Fig. 4 | Nanographene functionalization. a** Suzuki and nucleophilic aromatic substitution reactions for scaffold functionalization. **b** Chemical structures of modified nanographenes.

signals in the area of 6–10.5 ppm indicating a diverse reaction mixture (Supplementary Fig. 7).

## Nanographene functionalization

The regioselective chlorination offered an opportunity to functionalize the nanographene scaffold. For this, initially, a palladium-catalyzed Suzuki reaction with boronic acid **19** was carried out and afforded the product **20** in good isolated yield (41%) (Fig. 4a)[56]. X-ray crystallography confirmed the structure of the bis-coupled product (Fig. 5a). Encouraged by these results, nucleophilic substitution reaction on **7** was targeted. Scott and co-workers have established the utility of such reactions on the corannulene scaffold to achieve the synthesis of

thioethers[57]. By using thiocresol as the nucleophile, the synthesis of bis-thioether derivative **21** was achieved in an isolated yield of 72%. In this way, the regioselective chlorination allows for further chemical modification of the nanographene structure.

During the purification of **20**, trace side-products **22** and **23** were obtained which are likely to be formed through dehalogenation reaction (Fig. 4b). The isolation of these side-products was useful as it provided the fundamental corannulene-coronene hybrid and the intermediate mono-phenyl structure for comparing the structure-property relationship in the series **23**, **22**, and **20** (Supplementary Fig. 8 and Table 1). Finally, since the Suzuki reaction has a broader scope, all the chloride substrates (**11**, **12**, **16**, **18**) were subjected to coupling with

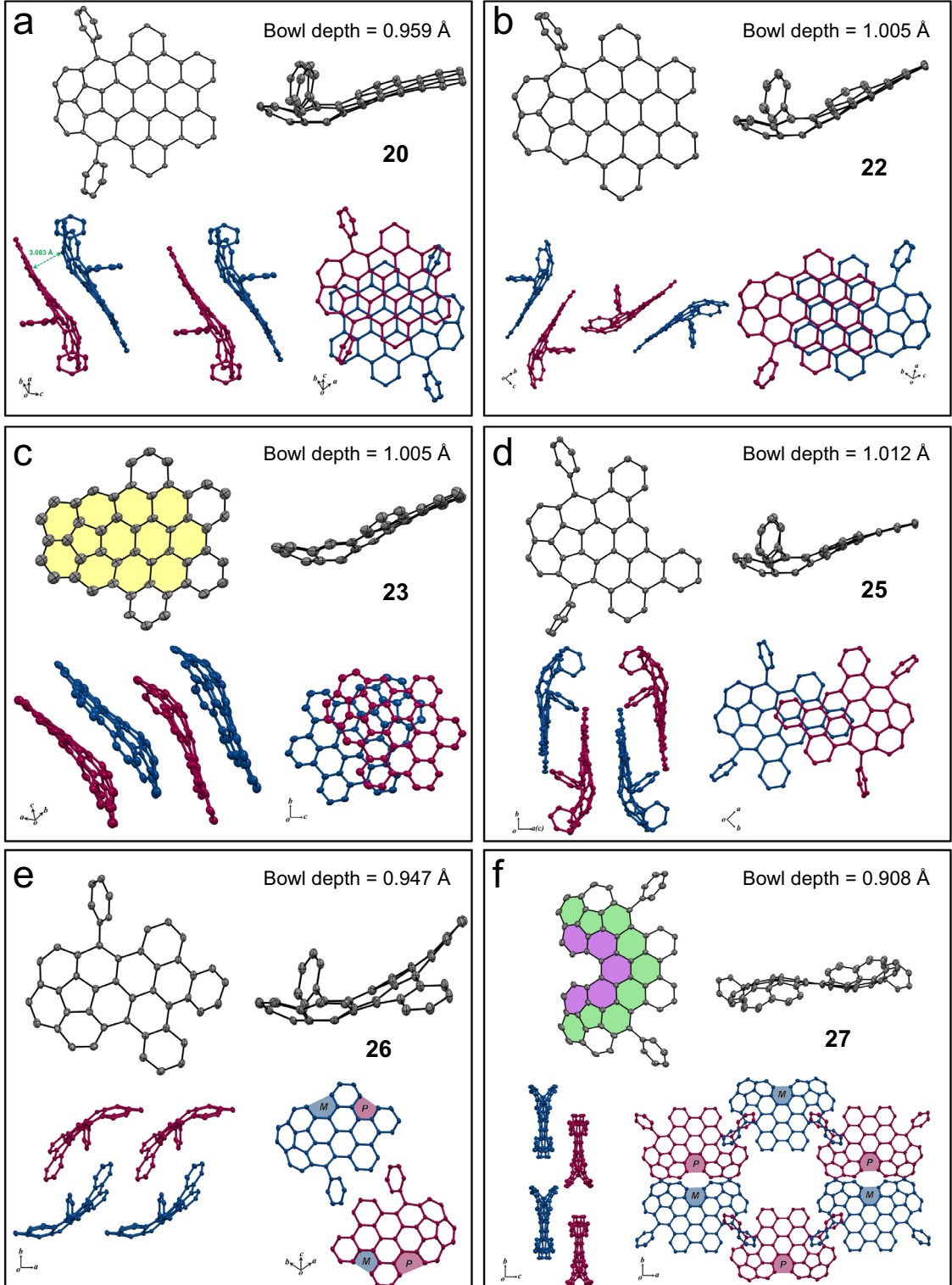

**Fig. 5 | X-Ray crystal structures of nanographenes.** Thermal ellipsoids were scaled at 50% probability level. *tert*-Butyl groups and hydrogen atoms were omitted for clarity. Top view, side view, and packing structures are shown in a clockwise fashion for **a 20**, **b 22**, **c 23** (the corannulene/coronene hybrid is highlighted), **d 25**, **e 26**, and **f 27** (the superhelicene structure is highlighted).

the boronic acid partner **19** to give expected products **24**, **25**, **26**, and **27** (Figs. 4b and 5b–e, and Supplementary Fig. 9). X-ray crystallographic analyses indicated the bowl depth to be relatively deeper for the nanographenes (e.g. 1.00 Å for **23**) than the parental bowl motif corannulene (0.87 Å) (Fig. 5 and Supplementary Table 1)[58]. Compounds **18** and **27** are unique as they are fully π-extended−super – [5]helicene structures mentioned before (Fig. 5f).

## Optical properties

In UV-Vis spectroscopy, corannulene and coronene display broad absorption bands below 400 nm[59,60]. In contrast, the nanographenes display several moderate-intensity absorption bands in the 400–550 nm region (Fig. 6a and Supplementary Figs. 10–22). This 150 nm bathochromic shift is a reflection of the extended π-conjugation in the nanographenes. The most red-shifted band is

**Table 1 | Properties of curved nanographenes**

| Nanographene | $\lambda_{onset}$ (nm) | $E_g^{opt}$ (eV)[a] | $\lambda_{em}$ (nm) | $E_{red,1}^{onset}$ (eV) | $E_{red,1}^{swv}$ (eV) | $E_{LUMO}$ (eV)[b] | $E_{HOMO}$ (eV)[c] |
|---|---|---|---|---|---|---|---|
| 7 | 523 | 2.37 | 522 | −1.72 | −1.78 | −3.08 | −5.45 |
| 11 | 482 | 2.57 | 479 | −1.87 | −1.96 | −2.93 | −5.50 |
| 12 | 500 | 2.48 | 505 | −1.80 | −1.86 | −3.00 | −5.48 |
| 16 | 477 | 2.60 | 484 | −1.88 | −1.98 | −2.92 | −5.52 |
| 18 | 493 | 2.52 | 494 | −1.72 | −1.84 | −3.08 | −5.60 |
| 20 | 518 | 2.39 | 518 | −1.89 | −2.00 | −2.91 | −5.30 |
| 21 | 541 | 2.29 | 546 | −1.69 | −1.79 | −3.11 | −5.40 |
| 22 | 510 | 2.43 | 520 | −1.84 | −1.96 | −2.96 | −5.39 |
| 23 | 504 | 2.46 | 517 | −1.87 | −1.94 | −2.93 | −5.39 |
| 24 | 476 | 2.60 | 488 | −2.01 | −2.09 | −2.79 | −5.39 |
| 25 | 499 | 2.48 | 515 | −1.93 | −2.01 | −2.87 | −5.35 |
| 26 | 474 | 2.62 | 492 | −1.98 | −2.07 | −2.82 | −5.44 |
| 27 | 499 | 2.48 | 509 | −1.81 | −1.93 | −2.99 | −5.47 |

[a]$E_g^{opt} = 1240/\lambda_{onset}$ (eV); [b]$E_{LUMO} = -(4.8 - E_{red,1}^{onset})$ (eV); [c]$E_{HOMO} = E_{LUMO} - E_g^{opt}$.

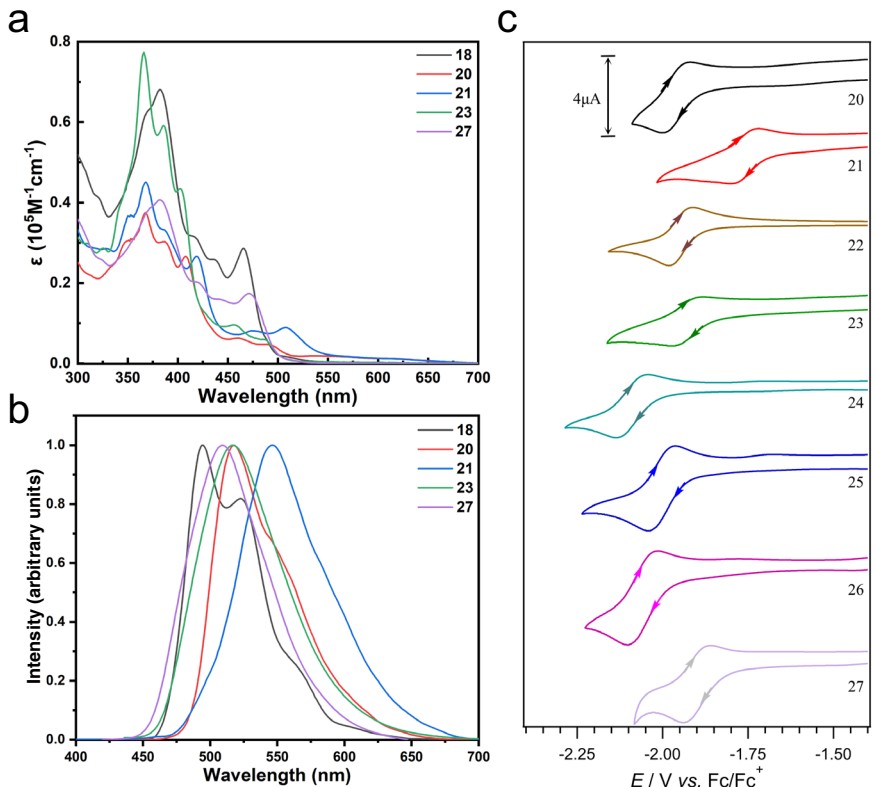

**Fig. 6 | Optical and electronic properties. a** UV-Vis absorption spectra in dichloromethane at room temperature at a concentration of $10^{-5}$ mol L$^{-1}$ (compounds **18**, **20**, **21**, **23**, and **27** are represented by black, red, blue, green, and purple colored lines, respectively). **b** Fluorescence emission spectra in dichloromethane at room temperature. The same color scheme is used as in the case of absorption spectra. **c** Cyclic voltammograms were measured at room temperature in dry, degassed dichloromethane with $n$-Bu$_4$NPF$_6$ (0.1 M) at a scan rate of 0.1 V s$^{-1}$ at a 1 mm diameter planar glassy carbon electrode under an argon atmosphere (nanographenes **20**, **21**, **22**, **23**, **24**, **25**, **26**, and **27** are represented by black, red, brown, green, teal, blue, magenta, and lavender colored lines, respectively).

observed at 510 nm for the thioether-functionalized nanographene **21**. The electron-donating nature of the sulfur atoms in **21** can rationalize the bathochromic shift and indicates an opportunity for varying the band-gap through post-synthesis functionalization approach. In **20** and **27**, the absorption bands are red-shifted when compared to **23** and **18**, respectively. Furthermore, both bis-functionalized nanographenes, **20** and **21**, display tailing of absorption until 650 nm (Supplementary Figs. 8 and 15–16). These results indicate that the (thio)phenyl rings can electronically communicate with the main aromatic scaffold and

underline the importance of functionalization in tuning the optical properties. The use of phenyl groups carrying electron donors (such as *N,N*-dimethylaniline) is expected to lead to a stronger bathochromic shift. In fluorescence emission spectroscopy, the nanographenes are emissive in the blue-green region of the electromagnetic spectrum (Fig. 6b and Supplementary Figs. 10–22).

Time-dependent (TD) Density Functional Theory (DFT) calculations were carried out on the model compounds **7′** and **18′** (where the *t*-Bu groups were replaced by methyl groups) to determine the nature of

the electronic transitions associated with the observed UV/Vis absorptions located at 500–550 nm. Our TD-DFT calculations (TD-B3LYP-D3/def2-SVP level) reproduce the occurrence of an electronic transition at ca. 500 nm ($\lambda_{calc}$ = 481 and 467 nm, for **7′** and **18′** respectively). This transition is the result of the one-electron promotion from HOMO to LUMO, which can be viewed as π-molecular orbitals delocalized along the entire system (Supplementary Fig. 23), therefore confirming the extension of the π-conjugation in the newly prepared nanographenes. Moreover, the HOMO→LUMO transition in **21** is significantly red-shifted ($l_{calc}$ = 540 nm) with respect to the analogous transition involving **7′** or **18′**, which is consistent with the experimental findings and a direct consequence of the involvement of the lone-pair of the sulfur atoms in the corresponding frontier molecular orbitals (Supplementary Fig. 23).

## Electrochemical properties

Würthner's curved imides[61–64] and Scott and Itami's warped nanographenes[65] represent elegant examples of electron-poor nanographenes. In this context, the electrochemical behavior of the synthesized nanographenes was studied with the help of square-wave voltammetry (SWV) and cyclic voltammetry (CV) in dichloromethane. These results indicated that all nanographenes are electron deficient molecules likely due to molecular curvature as is exemplified by the parental motif corannulene[66] (Fig. 6c and Supplementary Figs. 24–27). Overall, the first one-electron reduction occurs in the region of –1.8 to –2.0 V $vs$ Fc/Fc⁺ (Fc = ferrocene), 300–500 mV more easily than parental corannulene[66]. The chlorinated structures show limited chemical reversibility of the first reduction process, in the sense that the oxidative ($i_p^{ox}$) to reductive ($i_p^{red}$) peak current ratios ($i_p^{ox}/i_p^{red}$) were «1 (at a scan rate of 0.1 V s⁻¹) indicating chemical instability of the reduced compounds. However, as seen in modified cases for compounds **20**–**27** (Fig. 6c) whose CVs have $i_p^{ox}/i_p^{red}$-ratios that approach unity, functionalization is a means to improve the lifetime of the reduced compounds.

The nanographenes also display a second reduction process at potentials -200–400 mV more negative, with a similar current magnitude to the first reduction process, suggesting a second one-electron reduction to form the dianions. However, only compound **27** displays an $i_p^{ox}/i_p^{red}$-ratio close to unity for the second one-electron transfer process, indicating that only this nanographene forms a dianion that has a lifetime in solution of at least several seconds. Nanographene **27** is symmetrical and conjugated, thus the difference in potential between the first two one-electron reduction processes can be used to estimate the comproportionation equilibrium constant ($K_c$) for the internal electron transfer (Eq. 1)[67], where $E^0_1$ and $E^0_2$ are the formal potentials (that approximate the standard electrode potentials) of the first and second reduction processes, respectively, $F$ is the Faraday constant (96485 C mol⁻¹), $R$ is the gas constant (8.3143 J K⁻¹ mol⁻¹) and $T$ is the temperature (in K). Using a value of 0.2 V for the separation between the first two reduction processes for nanographene **27** leads to a $K_c$ value of approximation $2.4 \times 10^3$, indicating intermediate internal electronic interactions between the two symmetrical halves of the molecule.

$$K_c = \exp[(|E_1^0 - E_2^0|F)/RT] \qquad (1)$$

Overall, it is anticipated that electron-withdrawing substituents such as cyano or trifluoromethyl groups on the phenyl ring can further enhance the electron affinity of the nanographenes. Furthermore, large nanographenes such as **27** present some interesting materials for future electrochemical studies.

In summary, alkyne-carbonyl metathesis provides a corannulene-based indanone molecule which paves the way to a cyclopentadienone moiety useful in a Diels-Alder reaction with a variety of alkynes and arynes to give graphitization precursors. An iron(III) chloride-mediated mechanochemical Scholl reaction then annulates the Diels-Alder adducts to the corannulene nucleus. Interestingly the graphitization process activates the adjacent positions for a chlorination reaction. Thus, chlorinated curved nanographenes are obtained after the graphitization step. The isolated chemical yields for all the involved chemical processes are >60%. Each reaction is carried out multiple times and provides isolated yields within a similar range. Therefore, overall, the developed synthetic route can be described as general, efficient, and reproducible. Our DFT calculations indicate that the regioselectivity of the chlorination reaction can be rationalized by means of the condensed Fukui functions, which indicate that the edge corannulene rings bear the most reactive positions. In addition, it is found that aromaticity also plays a role in controlling regioselectivity. The regioselective chlorination offers an opportunity to functionalize the nanographene scaffold in an atomically precise manner. This could be achieved with the help of a boronic acid or a thiol molecule. Such functionalization strategy which efficiently alters the chemical structure and material properties can be potentially harnessed to prepare much stronger electron acceptors by placing electron-withdrawing groups on nanographenes. Alternatively, the placement of electron donating groups is expected to lead to red-absorbing/emitting nanographenes. A future goal would also be to render the nanographenes soluble in water for biological applications by placing polyethylene glycol (PEG) chains through the use of commercially available thiol-end-functionalized PEGs. It is anticipated that such constructs would be of interest as bio-relevant radical scavengers and antioxidants due to the electron-deficient nature of the curved aromatic scaffold.

## Data availability

All data supporting the findings of this study are available within the article and Supplementary information files, and also are available from the corresponding author upon request. The X-ray crystallographic coordinates for structures reported in this study have been deposited at the Cambridge Crystallographic Data Centre (CCDC), under deposition numbers CCDC 2202286 (compound **4**), CCDC 2202106 (compound **6**), CCDC 2234615 (compound **7**), CCDC 2202097 (compound **10**), CCDC 2202098 (compound **15**), CCDC 2202099 (compound **17**), CCDC 2202100 (compound **20**), CCDC 2202101 (compound **22**), CCDC 2202102 (compound **24**), CCDC 2202103 (compound **25**), CCDC 2202104 (compound **26**), and CCDC 2202105 (compound **27**). These data can be obtained free of charge from The Cambridge Crystallographic Data Centre via www.ccdc.cam.ac.uk/data_request/cif. Supplementary Data 1 contains the cartesian coordinates of the structures.

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

## Acknowledgements

Financial support from the Ministry of Education Singapore under the AcRF Tier 1 (MOE T1 RG11/21) and AcRF Tier 2 (MOE-T2EP10221-0002), M.C.S., and the Spanish MCIN/AEI/10.13039/501100011033 (Grants PID2019-106184GB-I00 and RED2018-102387-T), I.F., are gratefully acknowledged.

## Author contributions

J.S., R.W., Z.Z., and J.Z. carried out the experimental work and formulated data figures. M.C.S. designed the synthesis and supervised the progress of the project. I.F. carried out all the computational studies and wrote the theoretical part of the manuscript. R.D.W. designed and described all the electrochemical studies. I.F., R.D.W., and M.C.S. prepared the manuscript and revised it with the help of all the co-authors.

## Competing interests

The authors declare no competing interests.
