## [Peer Review File · Nature Communications]

Synthesis of Precisely Functionalizable Curved Nanographenes via Graphitization-Induced Regioselective Chlorination in a Mechanochemical Scholl ReactionREVIEWER COMMENTS

Reviewer #1 (Remarks to the Author):

Stuparu and coworkers describe an elegant application of the mechanochemical Scholl reaction that, in addition to the intended closing of a corannulene-based nanographene, attaches regioselectively additional chlorine atoms. The regioselectivity was tested on several structurally similar compounds with the same selectivity and the authors also use DFT to track the origin of regioselectivity. The reported reactions are high-yielding and the specific chlorination is highly valuable because there is no need for the usually impossible separation of chlorinated isomers and products with various degree of chlorination.

Overall, I find the work to be important and conducted in an impressively thorough manner. The manuscript is clearly written and composed.

I will be happy to recommend publication of this manuscript, but before I do, I'd like to see some details about X-ray analysis explained. Fig 1. depicts crystal structure of compound 7, but I do not see that this crystal structure has a CCDC deposition number assigned. Given that the authors extensively used X-ray analysis in characterisation, how come there are no crystallographic data on the chlorinated products of the Scholl reaction (compounds 11, 12, 16, 18)?

Reviewer #2 (Remarks to the Author):

Stuparu and coworker describe a synthesis pathway towards curved nanographenes including Cl-functionalization based on conventional and mechanochemical reaction steps. One characteristic is the regioselective chlorination at the edge corannulene rings. It should be acknowledged that a well-chosen substrate scope as well as a broad selection of characterization techniques have been applied to support the authors hypothesis. The work is very-well carried out and the argumentation is clear and comprehensible. Nevertheless, the novelty and the significance of the work may be ranked below the requirements for publishing with Nature Communications for the following reasons: 1) The authors have recently published pathbreaking articles in *Angewandte Chemie* (<https://doi.org/10.1002/ange.202007815>) and *Nature Communications* (<https://www.nature.com/articles/s41467-021-25495-6>) where they are reporting on curved nanographene synthesis via mechanochemical methods. Likewise, the Scholl reaction is a well-established tool for intramolecular C-C-bond formation in graphene structures and carried out and applied several times mechanochemically. Even, the Cl-functionalization/ chlorination of nanographenes by mechanochemistry has recently been reported by Baier et al. (<https://doi.org/10.1039/D1RA07679E>). I understand that the regioselectivity, which is described in this manuscript, adds to the already reported protocols, but I recommend a more specific journal for these findings.

As the curved-graphene synthesis itself has already been published several times by the authors, the focus should be more on the chlorination itself. I appreciate the DFT calculations included in this work, it would be of likewise great interest to carve out how reaction conditions influence this chlorination, e.g. Is the chlorination pattern a question of reaction time and FeCl₃ amount? What are the side products?

Reviewer #3 (Remarks to the Author):

The paper of Stanojkovic et al. demonstrates how starting from a corannulene-based indanone molecule, using Scholl reaction, one can obtain chlorinated derivatives which can be further used to make larger graphene-like structures. It is very interesting that the chlorinated derivatives, which are usually considered as side products, are obtained through reactions with a significant regioselectivity. I find this work very interesting and competently done. Anyway, I will restrict my consideration to the theoretical part of the manuscript. The authors used a standard methodology to assess the

aromaticity of the studied molecules. Namely, they employed NICS and ACID plots. It was shown that the aromaticity of the corannulene core is practically unchanged through all reactions in which the carbon skeleton is enlarged. This study also confirms that aromaticity of individual rings is preserved even in very curved and nonplanar systems. Along these lines, there are many papers dealing with this topic and the authors can consider some of these: Phys. Chem. Chem. Phys., 2011, 13, 16861–16866, J. Phys. Chem. A 2007, 111, 4513-4521, J. Phys. Chem. A 2017, 121, 3616-3626. The regioselectivity of the examined chlorination reactions was rationalized by means of the condensed Fukui functions. In addition, the TDDFT methods we used to analyze optical properties of the studied molecules.

In summary, the presented paper is well-written, well-structured, and thus I would recommend acceptance of the manuscript.

Reviewer #4 (Remarks to the Author):

The paper manuscript by Stanojkovic et al. describes a solid state cascade cyclization/chlorination reaction of a series of corannulene-based polycyclic aromatic hydrocarbons (PAHs), which they categorize as a synthetic chemists' version of "graphitization." The novelty of their work lies on the conceptual development on selective cascade aromatic cyclization (graphitization) and practical application of solid state Scholl reaction for large non-planar PAHs. The study is conducted thoroughly and the manuscript is well-written. This work will be of interest for a wide range of researchers in the field of organic synthetic chemistry and materials scientists looking for large nano carbon materials. Crystallographic analysis has been performed for a large portion of reported final products, which has largely been done properly and substantiates unambiguously the structures of these synthesized molecules. I recommend publication of this manuscript in nature communications after following minor concerns are addressed.

1. In Fig. 2. Why does not the [5]helicene moiety of 18 react to form an additional six-membered ring? Such a reaction would also form an antiaromatic six-membered ring, which should fit to the criteria described in page 9.
2. On page 8, the sentence 'Furthermore, in the bis-corannulene system, although the position of the annulated sites differs from that of nanographene 7, the bis-chlorination occurs with fidelity.' was not clear to me. In which sense the position of the annulated sites differs from that of nanographene 7?
3. On page 8 the authors describe the correlation of Fukui functions and the position of chlorination. Can the authors use the same functions to explain the reactivity of Scholl reaction? How does the plot shown in Fig 3a for their precursors (6, 9, 10, 15, and 17) look like?
4. On page 8, compound 4' is not defined.
5. On page 11, 1 Å should be 1.0 Å (or 1.00 Å depending on the accuracy).
6. On page 15, Wurthner should be Würthner. The same issues are found in reference list (i.e. Mullen should be Müllen, Stepien should be Stepien). Authors are asked to check the whole reference list to correct these issues.

Issues on crystallographic data

There are several issues related to the crystallographic data. The authors should consult a crystallographer and let all CIFs checked by them. Followings are the problems that I realized.

1. For all structures, how the structure was solved is not indicated. The authors should fill in `_atom_sites_solution_primary` in the CIFs.
2. M10: for this structure the hkl file is missing. It is required to embed original hkl data for CIF files and this should be done for this CIF.
3. M17: 1) two protons are missing at C98 and C99. 2) there seems to be an unsolved disorder around the dichloroethane (C11 C12 C125 C126) with a high electron density peak above 1.3 e.Å⁻³.
4. M20: Information on cell determination, crystal color, crystal shape is missing.
5. M22: A high second weight parameter (74.51) is used. The possibility of twinning should be examined.

Reviewer #1:

Stuparu and coworkers describe an elegant application of the mechanochemical Scholl reaction that, in addition to the intended closing of a corannulene-based nanographene, attaches regioselectively additional chlorine atoms. The regioselectivity was tested on several structurally similar compounds with the same selectivity and the authors also use DFT to track the origin of regioselectivity. The reported reactions are high-yielding and the specific chlorination is highly valuable because there is no need for the usually impossible separation of chlorinated isomers and products with various degree of chlorination. Overall, I find the work to be important and conducted in an impressively thorough manner. The manuscript is clearly written and composed. I will be happy to recommend publication of this manuscript, but before I do, I'd like to see some details about X-ray analysis explained. Fig 1. depicts crystal structure of compound 7, but I do not see that this crystal structure has a CCDC deposition number assigned. Given that the authors extensively used X-ray analysis in characterisation, how come there are no crystallographic data on the chlorinated products of the Scholl reaction (compounds 11, 12, 16, 18)?

Author Response: We are much indebted to the reviewer for the encouraging comments and for his/her time in reading and evaluating our manuscript. We appreciate it very much. We invested almost a year in efforts to crystalize the synthesized compounds. Under a number of conditions, chlorinated compounds did not yield crystals suitable for single-crystal analysis. The crystals are small and their quality is not satisfactory for the single-crystal analysis. Only after substitution of the chlorine atoms with the phenyl group, the compounds yielded suitable crystals for X-ray analysis. The only exception is chlorinated compound 7. Its CCDC deposition number is 2234615. This information is now included in the revised manuscript. Thank you.

Reviewer #2:

Stuparu and coworker describe a synthesis pathway towards curved nanographenes including Cl-functionalization based on conventional and mechanochemical reaction steps. One characteristic is the regioselective chlorination at the edge corannulene rings. It should be acknowledged that a well-chosen substrate scope as well as a broad selection of characterization techniques have been applied to support the authors

hypothesis. The work is very-well carried out and the argumentation is clear and comprehensible. Nevertheless, the novelty and the significance of the work may be ranked below the requirements for publishing with Nature Communications for the following reasons: 1) The authors have recently published pathbreaking articles in *Angewandte Chemie* (<https://doi.org/10.1002/ange.202007815>) and *Nature Communications* (<https://www.nature.com/articles/s41467-021-25495-6>) where they are reporting on curved nanographene synthesis via mechanochemical methods. Likewise, the Scholl reaction is a well-established tool for intramolecular C-C-bond formation in graphene structures and carried out and applied several times mechanochemically. Even, the Cl-functionalization/ chlorination of nanographenes by mechanochemistry has recently been reported by Baier et al. (<https://doi.org/10.1039/D1RA07679E>). I understand that the regioselectivity, which is described in this manuscript, adds to the already reported protocols, but I recommend a more specific journal for these findings. As the curved-graphene synthesis itself has already been published several times by the authors, the focus should be more on the chlorination itself. I appreciate the DFT calculations included in this work, it would be of likewise great interest to carve out how reaction conditions influence this chlorination, e.g. Is the chlorination pattern a question of reaction time and FeCl₃ amount? What are the side products?

Author Response: Let's establish the relevance of the three articles mentioned by the referee to the present work. The first article finds itself in a general pool of citations (39-42) on mechanochemical Scholl reaction. The second article is not even in the reference list because it is not relevant to the discussion. And, the third article is the antithesis of the present work (as described in the manuscript – please see discussion on page 6).

Let's now consider some elements of novelty in the current work. (1) Activation of specific and multiple positions on aromatic scaffolds in an efficient and general manner for in-situ halogenation is the first finding of its kind in a century old literature on Scholl chemistry. (2) General strategies for post-synthesis chemically precise functionalizations of nanographenes remain extremely rare to date. (3) All the nanographene structures are unique. For instance, the example of a laterally-fully-aromatically-extended helicene (helicene embedded in a continuous π -envelope) is only the 3rd known type of this superhelicene family. The helicene chemistry can also be considered ancient. To write new chapters in the century old chemistries and to

bring the new chemistry of nanographenes forward by showing that one can alter the chemical composition with atomic precision and thus the properties (and potentially applications), we believe, is suitable for the pages of *Nat. Commun.*

The non-planer scaffold is activated upon graphitization for mono/bis-chlorination. This is the essence of the manuscript. Once such mono/bis-chlorination occurs (depending upon the number of annulated sites), no further chlorinations occur at any position at the nanographene structure. This has been established with the help of many examples in the manuscript. In a Scholl reaction, typically, a large excess of FeCl₃ is used (>70 equivalents in the present work). Furthermore, the reaction time of 1-1.5 hours used in this work is sufficiently long for the (random) multi-chlorination to occur. Yet, only, regioselective mon/bis-chlorination based upon graphitization-induced activation sites is observed in high yields. Finally, side-products are not investigated because the isolated yields of the regioselective chlorination products are high.

Reviewer #3:

The paper of Stanojkovic et al. demonstrates how starting from a corannulene-based indanone molecule, using Scholl reaction, one can obtain chlorinated derivatives which can be further used to make larger graphene-like structures. It is very interesting that the chlorinated derivatives, which are usually considered as side products, are obtained through reactions with a significant regioselectivity. I find this work very interesting and competently done. Anyway, I will restrict my consideration to the theoretical part of the manuscript. The authors used a standard methodology to assess the aromaticity of the studied molecules. Namely, they employed NICS and ACID plots. It was shown that the aromaticity of the corannulene core is practically unchanged through all reactions in which the carbon skeleton is enlarged. This study also confirms that aromaticity of individual rings is preserved even in very curved and nonplanar systems. Along these lines, there are many papers dealing with this topic and the authors can consider some of these: *Phys. Chem. Chem. Phys.*, 2011, 13, 16861–16866, *J. Phys. Chem. A* 2007, 111, 4513-4521, *J. Phys. Chem. A* 2017, 121, 3616-3626. The regioselectivity of the examined chlorination reactions was rationalized by means of the condensed Fukui functions. In addition, the TDDFT methods we used to analyze optical properties of the studied molecules.

In summary, the presented paper is well-written, well-structured, and thus I would recommend acceptance of the manuscript.

Author Response: We would like to thank the reviewer for his/her time in reading and evaluating the manuscript. We appreciate it very much. Indeed. The referred works are related to the theoretical discussion in the manuscript and now included as references 50-52 in the revised manuscript. Thank you very much for pointing them out.

Reviewer #4:

The paper manuscript by Stanojkovic et al. describes a solid state cascade cyclization/chlorination reaction of a series of corannulene-based polycyclic aromatic hydrocarbons (PAHs), which they categorize as a synthetic chemists' version of "graphitization." The novelty of their work lies on the conceptual development on selective cascade aromatic cyclization (graphitization) and practical application of solid state Scholl reaction for large non-planar PAHs. The study is conducted thoroughly and the manuscript is well-written. This work will be of interest for a wide range of researchers in the field of organic synthetic chemistry and materials scientists looking for large nano carbon materials. Crystallographic analysis has been performed for a large portion of reported final products, which has largely been done properly and substantiates unambiguously the structures of these synthesized molecules. I recommend publication of this manuscript in nature communications after following minor concerns are addressed.

Author Response: We are sincerely grateful to the reviewer for all the comments, as they have helped us immensely in revising the manuscript. Thank you for all the time and efforts. Much appreciated!

1. In Fig. 2. Why does not the [5]helicene moiety of 18 react to form an additional six-membered ring? Such a reaction would also form an antiaromatic six-membered ring, which should fit to the criteria described in page 9.

Author Response: The barrier to transformation of [5]helicene into benzoperylene is likely due to the helical conformation. Such transformations are known to occur under photoirradiation conditions. Thermally, typically, benzoperylenes are accessed by the Diels-Alder cycloaddition to the bay region of perylene. We did not observe this transformation under mechanochemical conditions.

2. On page 8, the sentence ‘Furthermore, in the bis-corannulene system, although the position of the annulated sites differs from that of nanographene 7, the bis-chlorination occurs with fidelity.’ was not clear to me. In which sense the position of the annulated sites differs from that of nanographene 7?

Author Response: We meant to say that although the bis-chlorination occurs in this system too, however, the positions of chlorination are different than in 7. On reflection, the sentence does appear unclear and therefore we have removed it from the manuscript in response to this comment.

3. On page 8 the authors describe the correlation of Fukui functions and the position of chlorination. Can the authors use the same functions to explain the reactivity of Scholl reaction? How does the plot shown in Fig 3a for their precursors (6, 9, 10, 15, and 17) look like?

Author Response: We have computed the Fukui functions for the precursors as suggested by the reviewer. As expected, it is spread over the corannulene nucleus. This data is attached to the revised submission as ‘Related Manuscript file’ (Title: Fukui_Function_Scholl_Precursors). The Scholl reaction is actually defined by the proximity of the phenyl ring(s), and driven by the formation of new six-membered aromatic ring(s). Fukui functions are not relevant in defining the Scholl reaction. Therefore, we have not included the new computational data in the revised manuscript.

4. On page 8, compound 4’ is not defined.

Author Response: Thank you for pointing out this mistake. It is now corrected by removing the dash.

5. On page 11, 1 Å should be 1.0 Å (or 1.00 Å depending on the accuracy).

Author Response: Thank you. The mistake is now corrected.

6. On page 15, Wurthner should be Würthner. The same issues are found in reference list (i.e. Mullen should be Müllen, Stepien should be Stępień). Authors are asked to check the whole reference list to correct these issues.

Author Response: Thank you for pointing these mistakes out. We have now revised the reference list in the light of this comment.

Issues on crystallographic data

There are several issues related to the crystallographic data. The authors should consult a crystallographer and let all CIFs checked by them. Followings are the problems that I realized.

1. For all structures, how the structure was solved is not indicated. The authors should fill in `_atom_sites_solution_primary` in the CIFs.

Author Response: Thank you for your suggestion. We have added the following information related to solving the crystal structures to the supporting information file (Page 2 under Materials and Methods), “Diffraction intensity data were measured either at 103 K with a Bruker Kappa diffractometer equipped with a CCD detector, employing either Mo K α ($\lambda = 0.71073 \text{ \AA}$) radiation, with the SMART suite of programs (SMART version 5.628; Bruker AXS Inc., Madison, WI, USA, 2001). Structural solution and refinement were carried out with the SHELXTL suite of programs (Sheldrick, G. M. University of Göttingen: Göttingen, Germany, 2014). The intensities were corrected for Lorentz and polarization effects. The non-hydrogen atoms were refined anisotropically. Hydrogen atoms were placed using AFIX instructions.

2. M10: for this structure the hkl file is missing. It is required to embed original hkl data for CIF files and this should be done for this CIF.

Author Response: Thank you for pointing this out. We have revised the cif file as suggested.

3. M17: 1) two protons are missing at C98 and C99. 2) there seems to be an unsolved disorder around the dichloroethane (C11 C12 C125 C126) with a high electron density peak above 1.3 e.\AA^{-3} .

Author Response: Thank you for pointing it out. We have updated the cif file for M17 in light of this comment.

4. M20: Information on cell determination, crystal color, crystal shape is missing.

Author Response: Thank you. We have updated the cif file for M20 in light of this comment.

5. M22: A high second weight parameter (74.51) is used. The possibility of twinning should be examined.

Author Response: Thank you for the suggestion. We have checked for the possibility for twinning but did not find it to be the case.

REVIEWERS' COMMENTS

Reviewer #4 (Remarks to the Author):

The authors have addressed all my concerns in a satisfactory manner in their revised version of manuscript.

There seems to be unsolved disorder for the newly uploaded crystal structure of 7 where the whole molecule stacking towards the opposite direction as the ones solved are seen in the residual electron density maps (the chloride atoms of these disordered molecules are indicated as large positive residual density up to $1.93 \text{ e}\cdot\text{\AA}^{-3}$). The occupancy of the minor disordered part seems to be low (~ 0.1) and thus the refinement of this disorder would only be stable with rather strong constrains/restraints. The authors should decide if they explain the issue with residual density as minor disordered part that is too weak to be solved or solve the whole disorder. This issue would not change the discussion of the main text and thus no further changes would be required in the main text. Additionally, the authors should mention in the method section of SI that they solved this structure as twins and with the use of SQUEEZE, with proper description of how they did such processes and appropriate citation of programs used.

Reviewer #4:

The authors have addressed all my concerns in a satisfactory manner in their revised version of manuscript. There seems to be unsolved disorder for the newly uploaded crystal structure of 7 where the whole molecule stacking towards the opposite direction as the ones solved are seen in the residual electron density maps (the chloride atoms of these disordered molecules are indicated as large positive residual density up to $1.93 \text{ e} \cdot \text{\AA}^{-3}$). The occupancy of the minor disordered part seems to be low (~ 0.1) and thus the refinement of this disorder would only be stable with rather strong constrains/restraints. The authors should decide if they explain the issue with residual density as minor disordered part that is too weak to be solved or solve the whole disorder. This issue would not change the discussion of the main text and thus no further changes would be required in the main text. Additionally, the authors should mention in the method section of SI that they solved this structure as twins and with the use of SQUEEZE, with proper description of how they did such processes and appropriate citation of programs used.

Author Response: We are once again thankful to the reviewer for helping us with her/his suggestions. In the crystal structure of compound 7, the large positive residue densities near chlorine atoms are due to the disordering of the whole molecule; the minor part of the disorder is, however, too weak to be solved. The crystal structure of 7 was actually solved without difficulties by Squeeze/Platon (A. L. Spek, Acta Cryst. (2015). C71, 9-18) and refined as a two-component non-merohedral twin with BASF parameter and HKLF 5 reflection file obtained by TwinRotMat routine of PLATON (A. L. Spek, Acta Cryst. (1990). A46, c34). We have now revised the supplementary information file and added this information under supplementary methods (page 2). Thank you once again for all the comments and suggestions.